# Regularized Molecular Conformation Fields

**Lihao Wang**[1,*] **Yi Zhou**[2], **Yiqun Wang**[2], **Xiaoqing Zheng**[1], **Xuanjing Huang**[1,3], **Hao Zhou**[4]

[1]Fudan University, [2]ByteDance AI Lab
[3]Shanghai Collaborative Innovation Center of Intelligent Visual Computing
[4]Institute for AI Industry Research (AIR), Tsinghua University
{wanglh19, zhengxq, xjhuang}@fudan.edu.cn
{zhouyi.naive, yiqun.wang}@bytedance.com, zhouhao@air.tsinghua.edu.cn

## Abstract

Predicting energetically favorable 3-dimensional conformations of organic molecules from molecular graph plays a fundamental role in computer-aided drug discovery research. However, effectively exploring the high-dimensional conformation space to identify (meta)stable conformers is anything but trivial. In this work, we introduce RMCF, a novel framework to generate a diverse set of low-energy molecular conformations through sampling from a regularized molecular conformation field. We develop a data-driven molecular segmentation algorithm to automatically partition each molecule into several structural building blocks to reduce the modeling degrees of freedom. Then, we employ a Markov Random Field to learn the joint probability distribution of fragment configurations and inter-fragment dihedral angles, which enables us to sample from different low-energy regions of a conformation space. Our model constantly outperforms state-of-the-art models for the conformation generation task on the GEOM-Drugs dataset. We attribute the success of RMCF to modeling in a regularized feature space and learning a global fragment configuration distribution for effective sampling. The proposed method could be generalized to deal with larger biomolecular systems.[2]

## 1 Introduction

The spatial arrangement of atoms within a molecule, also known as molecular conformation, determines the molecular physico-chemical properties, which plays an essential role in downstream computer-aided drug discovery tasks. However, the high-dimensional conformation space spanned by all atomic degrees of freedom (DoF) makes it a great challenge to identify the local minima of the associated *potential energy surface* (PES) of a molecule. Recently, many machine learning (ML) models have achieved remarkable success in molecular conformation generation tasks [Mansimov et al., 2019, Simm and Hernandez-Lobato, 2020, Xu et al., 2021a, Ganea et al., 2021, Xu et al., 2022], exhibiting orders of magnitude faster conformation prediction speed than traditional computational simulation approaches. This makes ML generative models a powerful tool for high-throughput screening in drug discovery, especially large molecules.

To fully exploit the power of ML in molecular conformation generation, we need to tackle a few key challenges. First, a molecule exhibits invariance under SE(3) transformations (i.e., translation and rotation) in 3-dimensional (3D) Euclidean space. In other words, each molecular conformation is uniquely defined up to rigid motion. Hence a single molecule can adopt an infinite set of possible poses. Second, molecules exhibit a variety of dynamics under ambient conditions (e.g., bond rotation, vibration, etc.), leading to a possibly complex PES landscape of high dimension. This makes

---

[*]This work was done when Lihao Wang was a research intern at Bytedance AI Lab.
[2]The code is available at https://github.com/leowang1217/RMCF.

36th Conference on Neural Information Processing Systems (NeurIPS 2022).

it particularly challenging for ML models to identify local minima within this space to generate energetically favorable conformations effectively.

Some recently developed ML models tried to solve these problems from different perspectives. For instance, GraphDG [Simm and Hernandez-Lobato, 2020] and CGCF [Xu et al., 2021a] used distance geometry as invariant features for molecular representation learning. The drawbacks of such representations are that these invariant variables are potentially redundant and exhibit mutual dependency, which may cause numerical instability during training. One can also design equivariant networks to hard-code the invariance/equivariance condition into the model [Satorras et al., 2021, Xu et al., 2022]. These models circumvent using intermediate invariant features, but instead directly learn from the spatial arrangement of atoms. However, some studies [Cohen et al., 2018, Li et al., 2021] have suggested that the specialized equivariant layers may cause some loss in the expressiveness of the neural network. On the other hand, GeoMol [Ganea et al., 2021] predicts a local structure for each atom, followed by assembling these atom-based building blocks to form the molecular conformation. These models have difficulty dealing with cyclic graphs (e.g., benzyl group) and may lead to unreasonable conformation predictions.

In the meantime, we realize that the high-dimensional PES could be described in a reduced basis by considering only a few significant DoF that contribute to the low-energy conformations. Specifically, consider the well-known example of the potential energy diagram of ethane molecule as shown in Figure 1. While the molecule exhibits a total of 18 DoF, we could capture the most significant dynamics using a single variable, i.e., the rotation w.r.t. the carbon-carbon bond, quantified by the H-C-C-H dihedral angle. Other DoF in the ethane molecule, e.g., the stretching and bending of other bonds, do not have a significant impact on the potential energy landscape, such that they can be treated as small perturbations to the molecular conformation within a potential well. In other words, to serve the purpose of effectively sampling local minima for low-energy conformations, we could reduce the dimension of the conformation space by tracking only a few significant DoF, as long as they adequately shape the corresponding energy landscape (e.g., the "$\omega$" shape in Figure 1). It then follows naturally to

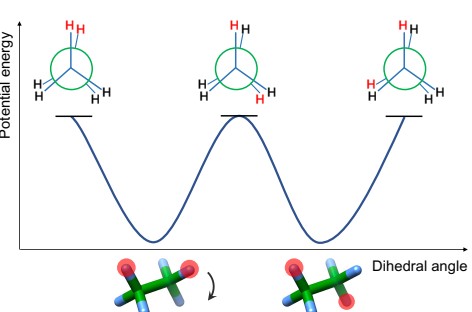

Figure 1: Schematic of the potential energy diagram of an ethane molecule. The upper panel shows the Newman projection of three degenerate eclipsed conformations, the lower panel shows two energetically-favorable staggered conformations. The H-C-C-H dihedral angle alone is sufficient to describe the potential energy change.

generate new conformers by sampling from the low-dimensional conformation space, which is also physically meaningful. In fact, a similar approach has been widely adopted by the molecular docking community, i.e., incremental construction [Meng et al., 2011], where a ligand molecule is partitioned into multiple fragments connected by rotatable covalent bonds. One may then gradually add ligand fragments with appropriate orientations to fit the ligand into the receptor pocket.

This paper presents Regularized Molecular Conformation Fields (RMCF), a novel framework for 3D molecular conformation generation. The novelty of this work lies in (1) developing a data-driven molecular segmentation algorithm to partition each molecule into fragments with low internal flexibility, which serves as a regularization term for our framework, and (2) employing a Markov Random Field (MRF [Murphy, 2012, Cotta et al., 2020]) to capture the joint probability distribution of fragment configurations and inter-fragment dihedral angles. Our work is partly inspired by the Ising model [Cipra, 1987] for simulating quantum spin systems, where we make an analogy between the spin state (i.e., spin up or down) and the configuration of each molecular fragment (e.g., chair or boat conformation of cyclohexane). The introduction of molecular fragments effectively reduces the dimension of the conformation space, where we only keep the most significant components (i.e., fragment configuration and inter-fragment dihedral angle) for conformation generation. This molecular segmentation serves as a regularization term for our framework, which is data-driven and has the effect of reducing feature dimensions. Then, we employ MRF as a generalized setting of the Ising model (with more spin states and more coupling terms) to capture the relationship between adjacent fragments and model the uncertainty of conformations. By estimating the parameters of the MRF, we can obtain a potential energy surface associated with different conformations. Therefore,

we can later sample from the low-energy region of PES, which is implicitly learned by the MRF, to obtain energetically favorable molecular conformations.

We demonstrate the effectiveness of RMCF using the GEOM-QM9 and GEOM-Drugs dataset[Axelrod and Gomez-Bombarelli, 2022], where results show that RMCF significantly outperforms state-of-the-art ML models. Specifically, our model can generate a diverse set of low-energy conformations located at distinct local minima of the underlying PES. We attribute the success of our model to the automatic construction of low-dimensional conformation space and learning a joint probability distribution using MRF to achieve effective sampling of new conformations. We argue that capturing the governing dynamics of molecular systems could significantly improve the performance of learning models for generative purposes.

## 2   Related Work

Recently, various machine learning models have been proposed for molecular conformation generation. CVGAE [Mansimov et al., 2019] first used a variational autoencoder (VAE) model to generate conformation with atomic coordinates. The model suffers from multi-modality problems due to invariance under SE(3) transformations. Some studies tried to address this problem using two main types of approaches. The first type of approach use intermediate invariant features, such as atomic distance, to encode the system, then leverage geometric algorithms to solve a set of atomic coordinates that match the invariant quantities. For example, GraphDG [Simm and Hernandez-Lobato, 2020] and CGCF [Xu et al., 2021a] proposed to predict the distance matrix by VAE and Flow, respectively, and solve the geometry through the Distance Geometry (DG) method [Liberti et al., 2014]. However, these invariant features are potentially redundant and could exhibit mutual dependency, which makes these methods numerically unstable and could predict unreasonable conformations. Some studies suggest that inconsistencies between training and test are responsible for the poor performance of the models [Shi et al., 2021, Xu et al., 2021b]. ConfGF [Shi et al., 2021] passed the gradient of loss to coordinates and ConfVAE [Xu et al., 2021b] used a bilevel optimization to alleviate the inconsistencies. GeoMol [Ganea et al., 2021] defined another set of invariants, which are bond lengths, bond angles, and dihedral angles. Zhu et al. [2022] proposed a model invariant to roto-translation of coordinates of conformations and permutation of symmetric atoms in molecules. Geomol generated multiple conformations by adding noise to the input, which did not learn a global energy function from which to sample the low-energy conformations.

Another type of approach applies equivariant networks or kernels, such that after the input is rotated and translated, the output will be transformed accordingly. Satorras et al. [2021] proposed an equivariant normalizing flow, E(n)-flow, and Xu et al. [2022] proposed GeoDiff, which is a diffusion model [Song and Ermon, 2019] with equivariant Markov kernels. Guan et al. [2021] proposed a variant of SE(3)-equivariant neural networks to learn the gradient fields of an implicit conformational energy landscape. Equivariant methods circumvent intermediate invariants and model the coordinates directly, but require the design of specialized equivariant layers, which some studies have suggested would lose the expressive power of the network [Li et al., 2021].

In addtion, some software for conformation generation are widely used in biochemical research. For instance, RDKit [Riniker and Landrum, 2015] is a popular open-source software which generates conformations using ETKDG distance geometry, and OMEGA is a commercial software which assembles the fragments with knowledge-based rules to generate conformations.

## 3   RMCF: Generating in Regularized Conformation Space

In this section, we present the regularized molecular conformation field (RMCF) in detail. Intuitively, RMCF partitions the original molecular graph into multiple molecular fragments, where adjacent fragments in the graph are connected with associated dihedral angles. In such a case, the molecular conformation generation problem can be decomposed into a) generating the conformation of molecular fragments, and b) determining the specific dihedral angles between connected fragments. The RMCF learns the potential energy landscape of molecular conformations by modeling a joint probability distribution of fragment conformation and dihedral angles, from which we can perform inference to draw diverse samples.

### 3.1 Overview of RMCF

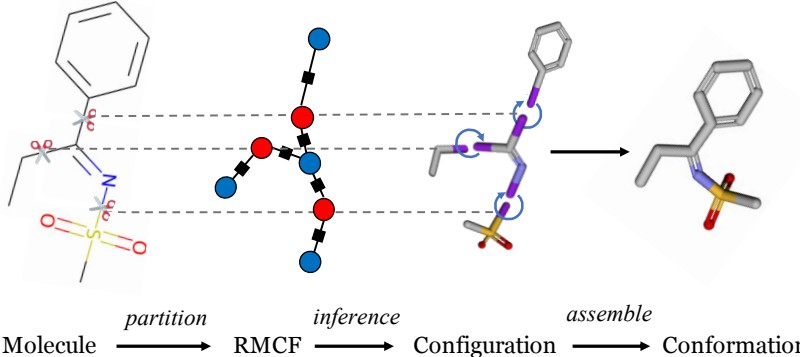

$$\text{Molecule} \xrightarrow{\textit{partition}} \text{RMCF} \xrightarrow{\textit{inference}} \text{Configuration} \xrightarrow{\textit{assemble}} \text{Conformation}$$

Figure 2: The workflow of RMCF. Starting from a 2D molecular graph, we partition the molecule into fragments with least intra-fragment DoF. Blue and red circles denote fragment and dihedral angle configurations, respectively, while the black squares denotes the interaction between neighboring configurations. We then use MRF to model the joint probability distribution of fragment and dihedral configurations. The last step is to assemble the predicted molecular conformation according to the predicted dihedral angles and fragment conformations.

We start with an overview of the our proposed RMCF model. Generally, as shown in Figure 2, given a molecular graph $G$, to obtain a generated 3D conformation $\mathcal{C}$, RMCF mainly takes three steps:

$$G \xrightarrow{1} \mathcal{G} \xrightarrow{2} \mathcal{X} \xrightarrow{3} \mathcal{C}$$

1. $G \longrightarrow \mathcal{G}$ (Section 3.2): The main idea of our proposed model is to build a conformation distribution within a regularized molecular conformation space, which is achieved by partitioning the molecular graph $G$ into multiple molecular fragments $F$ connected by dihedral angles $D$. Notably, the graph partition should satisfy the least intra-fragment DoF principle (introduced in Section 3.2), which is crucial for reducing the dimension of the molecular conformation space.

   The resulting graph after partitioning (including $F$ and $D$) could be jointly modeled with a Markov random field (also called undirected graphical model). Since it is regularized with reduced total DoF, we name our model Regularized Molecular Conformation Fields.

   Formally, a regularized conformation field $\mathcal{G} = (F, D, E)$ is an undirected graph formed by a collection of fragment vertices $F = (f_1, f_2, \cdots, f_{N_F})$, a collection of dihedral vertices $D = (d_1, d_2, \cdots, d_{N_D})$, and a collection of edges $E = (e_1, e_2, \cdots, e_{N_E}) \subset F \times D$ (edges between fragments and dihedral angles in the graph), where $N_F$, $N_D$, $N_E$ are the size of $F$, $D$, $E$, respectively [Wainwright et al., 2008]. Each edge consists of a pair of vertices $f \in F$ and $d \in D$. We associate with each vertex $f \in F$ a random variable $X_f$ taking values from a set of fragment conformations $\overline{\mathcal{C}_f}$ (see Section 3.2), each vertex $d \in D$ a random variable $X_d$, representing the value of dihedral angles.

2. $\mathcal{G} \longrightarrow \mathcal{X}$ (Section 3.3 and 3.4): Given $\mathcal{G}$, the conformation generation problem turns into generating the conformation of molecular fragments and determining the dihedral angles between them, namely inferring a configuration $\mathcal{X}$ from $\mathcal{G}$. A configuration $\mathcal{X} = (\mathcal{X}_f, \mathcal{X}_d)$ of the RMCF contains a set of fragment conformation $\mathcal{X}_f = \{x_{f_i} | i = 1, 2, \cdots, N_F\}$ and a set of dihedral angle values $\mathcal{X}_d = \{x_{d_i} | i = 1, 2, \cdots, N_D\}$, which could be assembled into the 3D conformation of a molecule. $\mathcal{X}$ could be obtained by maximizing the probability modeled by RMCF:

$$P(x_{f_1}, \cdots, x_{f_{N_F}}, x_{d_1}, \cdots, x_{d_{N_d}}) = \frac{1}{Z} \left\{ \prod_{f_i \in F} \psi(x_{f_i}) \cdot \prod_{d_i \in D} \psi(x_{d_i}) \cdot \prod_{(f_i, d_j) \in E} \psi(x_{f_i}, x_{d_j}) \right\} \quad (1)$$

where $Z$ is the normalizing factor, $\psi$ is the compatibility functions, which will be further described in Section 3.3.

3. $\mathcal{X} \longrightarrow \mathcal{C}$ (Section 3.5): Given the configuration $\mathcal{X}$, we assemble the predicted fragment conformation and dihedral angle values into the corresponding molecular conformation $\mathcal{C}$.

## 3.2 Graph Construction with Least DoF Principle

In this section, we elaborate more about the molecular graph partition with least DoF principle used in our proposed RMCF. This is a crucial step to our model since it builds the foundation of the following probabilistic modeling. The entire graph construction process has three main components, namely a) constructing the fragment library with BRICS, b) obtaining fragment conformation vocabulary with clustering, and c) yielding final fragment partition by minimizing the intra-fragment DoF.

Specifically,We use BRICS, a decomposition method based on molecular functional groups, to pre-cut all molecular graphs and their 3D conformations and collect each fragment conformation into a set $\mathcal{C}_f$[3]. A sufficient number (i.e., 1,000) of fragment conformations are sampled from $\mathcal{C}_f$ for the following clustering steps.

Then, we use the K-Medoids clustering algorithm and use the root-mean-square deviation (RMSD) of two fragment conformations as the distance metric between two fragments, which takes rotation and translation invariance into consideration. We adjust the number of cluster centers $k$ from 1 to 10 and use the Silhouette Coefficient[Rousseeuw, 1987] to measure the goodness of these clusters and choose the optimal hyper-parameter k with clusters having the highest Silhouette Coefficient value. Cluster centroid elements are collected into conformation vocabulary $\overline{\mathcal{C}_f}$.

Although BRICS presents a solution to partition the molecular graph, it cannot guarantee a small DoF within the fragments. Therefore, we define a metric to quantify the *intra*-fragment DoF and design an algorithm to re-fragment the molecules to minimize this quantity. We define the fragment DoF as the conformation variance $\mathrm{Var}(\mathcal{C}_f)$ which is the sum of the squares of root-mean-square deviation(RMSD) between fragment conformations $\mathcal{C}_f$ and their cluster centers $\overline{\mathcal{C}_f}$, and the total DoF of a partitioned molecule as the mean of its fragment DoF. The best partition strategy $\mathcal{P}^*$ should have the least DoF $\mathcal{F}(\mathcal{P})$:

$$\mathcal{F}(\mathcal{P}) = \frac{1}{|\mathcal{P}|} \sum_{f \in \mathcal{P}} \mathrm{Var}(\mathcal{C}_f) = \frac{1}{|\mathcal{P}|} \sum_{f \in \mathcal{P}} \frac{1}{|\mathcal{C}_f|} \sum_{\mathcal{C}_f} \mathrm{RMSD}(\mathcal{C}_f, \overline{\mathcal{C}_f})^2 \tag{2}$$

$$\mathcal{P}^* = \arg\min_{\mathcal{P}} \mathcal{F}(\mathcal{P}) \tag{3}$$

We could employ graph dynamic programming to search for the optimal solution and the detailed algorithm is in Appendix B.

After partition, we can get the fragment set $F = (f_1, f_2, \cdots, f_{N_F})$ and insert the dihedral vertices between connected fragments to establish the RMCF $\mathcal{G}$ on the molecular graph $G$ as shown in the Figure 2.

## 3.3 Approximated Training of RMCF

**Training Objective** Training in RMCF is intractable, especially when it contains loops [Murphy, 2012]. It is difficult to estimate the normalization factor $Z$ in the objective (Eq. 1),which means we have to marginalize all possible conformations. We approximate the likelihood through *piece-wise training*, which has been proven efficient in training graphical models [Sutton and McCallum, 2009, 2012, Lin et al., 2016, Qu et al., 2022]. The key idea is to train each node and edge independently.

$$\mathcal{L} = -\log P(x_{f_1}, \cdots, x_{f_{N_F}}, x_{d_1}, \cdots, x_{d_{N_d}})$$

$$\approx \sum_{f_i \in F} \underbrace{\boxed{-\log \frac{\psi(x_{f_i})}{Z_f}}}_{\triangleq \mathcal{L}_n(f)} + \sum_{d_i \in D} \underbrace{\boxed{-\log \frac{\psi(x_{d_i})}{Z_d}}}_{\triangleq \mathcal{L}_n(d)} + \sum_{(f_i, d_j) \in E} \underbrace{\boxed{-\log \frac{\psi(x_{f_i}, x_{d_j})}{Z_e}}}_{\triangleq \mathcal{L}_e(f,d)} \tag{4}$$

where $\mathcal{L}_n$ and $\mathcal{L}_e$ denote the node-wise and edge-wise negative log-likelihood. Especially, $\mathcal{L}_e(f, d)$ measures *how probable a fragment $f$ and a dihedral $d$ appear at the same time*.

More concretely, for a pair of nodes in the field $e = (f, d) \in \mathcal{G}$, let $|f|$ and $|g|$ denote the cardinality of the state space, $|\mathbf{h}|$ denotes the hidden dimension, $y_f$ and $y_d$ are the ground truth of fragment conformation $f$ and dihedral angle value $d$, which will introduced calculation in the next paragraph. We calculate the representations from the graph neural networks: Let $\mathbf{h}_f^0 = \mathrm{emb}(f)$ and $\mathbf{h}_d^0 = \mathrm{emb}(d)$

---

[3]On the base of BRICS, we further cut the links of rings and side chains to prevent the combinatorial explosion

denote the fragment embedding and the dihedral embedding (which is a "dummy" embedding) as the network input, we compute the contextualized fragment representations $\mathbf{h}_f^{\ell+1}$ and dihedral representations $\mathbf{h}_d^{\ell+1}$ by:

$$\mathbf{h}_f^{\ell+1} = g\left(\mathbf{h}_f^\ell, \left\{\mathbf{h}_d^\ell\right\}_{d \in \mathcal{N}_f}\right), \quad \mathbf{h}_d^{\ell+1} = g\left(\mathbf{h}_d^\ell, \left\{\mathbf{h}_f^\ell\right\}_{f \in \mathcal{N}_d}\right) \tag{5}$$

where $\ell$ denotes the layer index, $g$ denotes the massage-passing function of GNN, $\mathcal{N}_f$ and $\mathcal{N}_d$ denotes the neighborhoods of fragment and dihedral respectively. The node-wise negative log-likelihood is defined as:

$$\mathcal{L}_n(f) = -\log\operatorname{softmax}\left(\operatorname{out}(\mathbf{h}_f)\right)[y_f], \quad \mathcal{L}_n(d) = -\log\operatorname{softmax}\left(\operatorname{out}(\mathbf{h}_d)\right)[y_d] \tag{6}$$

where $\operatorname{out}(\cdot): \mathbb{R}^{|\mathbf{h}|} \to \mathbb{R}^{|f|}$ is an operator parameterized by a linear layer. The edge-wise negative log-likelihood is defined as:

$$\mathcal{L}_e(f, d) = -\log\operatorname{softmax}\left(\mathbf{E}_f \mathbf{W}_e \mathbf{E}_d^\mathrm{T}\right)[y_f, y_d] \tag{7}$$

where $\mathbf{E}_f \in \mathbb{R}^{|f| \times |\mathbf{h}|}$ and $\mathbf{E}_d \in \mathbb{R}^{|d| \times |\mathbf{h}|}$ are the representations of a set of node states, $\mathbf{W}_e \in \mathbb{R}^{|\mathbf{h}| \times |\mathbf{h}|}$ is a matrix carrying the global information between two nodes, parameterized by a neural network $\mathbf{W}_e = \operatorname{MLP}(\mathbf{h}_f, \mathbf{h}_d)$ [Sun et al., 2019].

**Obtaining ground truth**    The oracle configuration $\mathcal{Y} = (\mathcal{Y}_f, \mathcal{Y}_d)$ contains a set of ground truth fragment conformations indexes $\mathcal{Y}_f = \{y_{f_i} | i = 1, 2, \cdots, N_F\}$ and a set of dihedral angle indexes $\mathcal{Y}_d = \{y_{d_i} | i = 1, 2, \cdots, N_D\}$ denote the choices of fragments conformations and dihedral angles, respectively. For a given fragment $f$, we pick the configuration $s$ whose corresponding fragment conformation $\mathcal{C}_s$ in vocabulary $\overline{\mathcal{C}_f}$, has the minimal RMSD to the actual fragment conformation $\mathcal{C}$, as the ground truth $y_f$.

$$y_f = \arg\min_s \operatorname{RMSD}(\mathcal{C}, \mathcal{C}_s), \mathcal{C}_s \in \overline{\mathcal{C}_f} \tag{8}$$

The dihedral angle $\varphi$ is defined as the angle between half-planes of two connected 3D fragments. Since the dihedral angle depends on the set of the atoms involved in the calculation, for the same type of fragments, we should keep the same set of atoms to calculate.

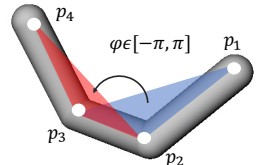

Figure 3: Dihedral angle

For a system where two fragments contain four consecutively-bonded atoms, two half-planes intersect on a rotatable bond. The angle between them is the *dihedral angle*. If the connected points are sequentially numbered and located at positions $\mathbf{p}_1, \mathbf{p}_2, \mathbf{p}_3, \mathbf{p}_4$ and the corresponding bond vectors are defined as $\mathbf{u}_1 = \mathbf{p}_2 - \mathbf{p}_1, \mathbf{u}_2 = \mathbf{p}_3 - \mathbf{p}_2, \mathbf{u}_3 = \mathbf{p}_4 - \mathbf{p}_3$. Then we have:

$$\varphi(\mathbf{u}_1, \mathbf{u}_2, \mathbf{u}_3) = \operatorname{atan2}\left(|\mathbf{u}_2| \, \mathbf{u}_1 \cdot (\mathbf{u}_2 \times \mathbf{u}_3), (\mathbf{u}_1 \times \mathbf{u}_2) \cdot (\mathbf{u}_2 \times \mathbf{u}_3)\right) \tag{9}$$

In practice, we quantize the dihedral angles $\varphi$ into a number of evenly divided bins which has interval length $L$ to discretize the continuous angle values

$$y_d = \lfloor \frac{180\varphi}{\pi L} \rfloor \tag{10}$$

### 3.4   Inference & Sampling

In this section, we give a brief overview of how we perform inference and sampling in RMCF.

**Inference**    If a RMCF is acyclic (linear- or tree-structured), the max-a-posterior decoding can be done via dynamic programming [Forney, 1973]. Cases are a bit complicated when loops occur in the RMCF. We adopt a more generalized algorithm, namely loopy belief propagation (LBP) [Murphy et al., 2013], to overcome such obstacles. We refer the reader to textbooks for more details about the LBP algorithm [Murphy, 2012]. It is worth noting that LBP is used only during inference time. We do not perform any inference during training due to the issue of training stability [Qu et al., 2022].

**Sampling**    Markov Chain Monte Carlo is popular in sampling from graphical models [Andrieu et al., 2003]. We employ a simple variant of it, i.e., Gibbs sampling [George and McCulloch, 1993], to draw

samples from RMCF. Specifically, we resample each fragment or dihedral individually, keeping all other nodes fixed in each iteration [Murphy, 2012]. The detailed algorithm can be found in Appendix C.

**Clustering** Once the Markov chain achieves detailed balance, we randomly draw $10,000$ samples from it. However, these samples can be governed by a few conformations, which means most of them are similar. We measure the distance between two configurations $\mathcal{X}^{(1)}$ and $\mathcal{X}^{(2)}$ by:

$$\mathrm{d}(\mathcal{X}^{(1)}, \mathcal{X}^{(2)}) = \|\mathcal{X}_d^{(1)} - \mathcal{X}_d^{(2)}\|_2^2 + \mathbb{H}(\mathcal{X}_f^{(1)}, \mathcal{X}_f^{(2)}) \tag{11}$$

where $\| \cdot \|_2$ is the Euclidean norm measuring the difference between dihedrals, $\mathbb{H}$ is the Hamming distance [Hamming, 1950] counting the number of different fragment configurations. We further run K-means clustering [MacQueen et al., 1967] based on the pair-wise distance to partition all samples into a fixed number of clusters and randomly sample an element from each cluster.

### 3.5 Assembling

Once we have predicted the dihedral angle $\hat{x}_d$ between two fragments, we need to compute the transformation matrix $\mathbf{T}$ to align the two fragments in arbitrary position and make their dihedral angle equal to the predicted angle and repeat these steps until the whole conformation is assembled.

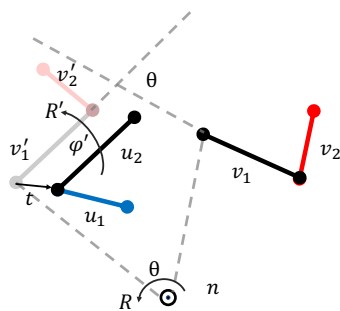

Given two fragments with six points, located at $\mathbf{p}_1, \mathbf{p}_2, \mathbf{p}_3, \mathbf{p}_4,$ $\mathbf{p}_5, \mathbf{p}_6$, the bond vectors are $\mathbf{u}_1 = \mathbf{p}_2 - \mathbf{p}_1, \mathbf{u}_2 = \mathbf{p}_3 - \mathbf{p}_2, \mathbf{v}_1 = \mathbf{p}_5 - \mathbf{p}_4$, $\mathbf{v}_2 = \mathbf{p}_6 - \mathbf{p}_5$. $\mathbf{u}_2$ and $\mathbf{v}_1$ are the same bond after assembling, so we have $|\mathbf{u}_2| = |\mathbf{v}_1|$.

Figure 4: Transformations for Assembling

First we need to rotate the second fragment around some axis so that $\mathbf{u}_2$ and $\mathbf{v}_1$ will be parallel. This axis is the unit normal vector $\mathbf{n}$ towards the plane of $\mathbf{u}_2$ and $\mathbf{v}_1$, and the rotate angle is equal to the cross angle $\theta$.

$$\cos\theta = \frac{\mathbf{u}_2 \cdot \mathbf{v}_1}{|\mathbf{u}_2| \, |\mathbf{v}_1|}, \quad \mathbf{n} = \frac{\mathbf{u}_2 \times \mathbf{v}_1}{|\mathbf{u}_2 \times \mathbf{v}_1|} \tag{12}$$

By using the Rodrigues' rotation formula, we can obtain the rotation matrix with unit normal vector $\mathbf{n}$ and rotation angle $\theta$. Next, we rotate the second fragment around $\mathbf{u}_2$ to match with the target dihedral angle. Finally we align the anchor points by calculating the final transformation matrix $\mathbf{T}$ as a composite matrix of two rotations and one translation.

$$\mathbf{R}(\mathbf{n}, \theta) = I\cos\theta + (1-\cos\theta)\begin{pmatrix} n_x \\ n_y \\ n_z \end{pmatrix}(n_x, n_y, n_z) + \sin\theta\begin{pmatrix} 0 & -n_z & n_y \\ n_z & 0 & -n_x \\ -n_y & n_x & 0 \end{pmatrix} \tag{13}$$

$$\mathbf{R}' = \mathbf{R}(\mathbf{u}_2, \frac{\pi L \hat{x}_d}{180} - \varphi(\mathbf{u}_1, \mathbf{u}_2, \mathbf{R}(\mathbf{n}, \theta)\mathbf{v}_2)), \quad \mathbf{t} = (\mathbf{p}_1 - \mathbf{R}'\mathbf{R}\mathbf{p}_4)^{\mathbf{T}} \tag{14}$$

$$\mathbf{T} = \begin{pmatrix} \mathbf{R}'\mathbf{R} & \mathbf{t} \\ \mathbf{0}^{\mathrm{T}} & 1 \end{pmatrix} \tag{15}$$

## 4  Experiment

We now demonstrate the effectiveness of RMCF on the conformation generation task for drug-like molecules.

### 4.1  Dataset and Baseline

Following previous work on conformation generation, we benchmark our model performance using the GEOM-QM9 and GEOM-Drugs dataset, which contains small and mid-sized organic molecules with high quality conformations. Considering the small size of molecules in QM9, we put the model performance in Appendix D, and mainly focus on discussing GEOM-Drugs results hereafter. We use the same test set as that in GeoDiff, where the remaining molecules are used for training and validation with a 9:1 ratio. The final training/validation/test set contains 271,539/30,171/1,034 molecules, respectively.

Our molecular segmentation algorithm eventually returned a vocabulary of 9,081 types of 2D fragments and 30,408 types of 3D fragments for the GEOM-Drugs dataset. As for the angular discretization of dihedral angles, the 360 degree interval is evenly divided into 72 bins. We adopt the Message-Passing Neural Network (MPNN) [Gilmer et al., 2017] as the framework for implementing our graph neural network. Other implementation details are provided in Appendix A.

## 4.2 Evaluation Metrics

To quantitatively measure the quality and diversity of generated molecular conformations, we use the same evaluation metrics as defined in Ganea et al. [2021] and Xu et al. [2022]. Let $S_g$ and $S_r$ denote the set of generated and reference conformations, respectively. The coverage score (COV) and matching score (MAT) following the conventional Recall measurement can be defined as:

$$\text{COV-R}\left(S_g, S_r\right) = \frac{1}{|S_r|} \left| \left\{ \mathcal{C} \in S_r \mid \text{RMSD}(\mathcal{C}, \hat{\mathcal{C}}) \leq \delta, \hat{\mathcal{C}} \in S_g \right\} \right|$$

$$\text{MAT-R}\left(S_g, S_r\right) = \frac{1}{|S_r|} \sum_{\mathcal{C} \in S_r} \min_{\hat{\mathcal{C}} \in S_g} \text{RMSD}(\mathcal{C}, \hat{\mathcal{C}})$$

(16)

where the threshold $\delta$ in coverage score is set as $1.25$Å for the GEOM-Drugs dataset in our work. The corresponding prediction precision metrics, i.e., COV-P and MAT-P, are defined in a similar manner, where the set of generated and reference conformations are swapped in the above definition. We set $S_g$ as twice the size of $S_r$ for each molecule for fair comparison with previous work [Ganea et al., 2021, Xu et al., 2022]. The precision-associated metrics focus more on generating accurate conformations that match those in the reference dataset, while the recall-associated metrics emphasize the structural diversity of generated conformations.

Table 1: Results on the **GEOM-Drugs** dataset, without FF optimization.

| Models | COV-R (%) ↑ | | MAT-R (Å) ↓ | | COV-P (%) ↑ | | MAT-P (Å) ↓ | |
|---|---|---|---|---|---|---|---|---|
| | Mean | Median | Mean | Median | Mean | Median | Mean | Median |
| CVGAE | 0.00 | 0.00 | 3.070 | 2.994 | - | - | - | - |
| GraphDG | 8.27 | 0.00 | 1.972 | 1.985 | 2.08 | 0.00 | 2.434 | 2.410 |
| CGCF | 53.96 | 57.06 | 1.249 | 1.225 | 21.68 | 13.72 | 1.857 | 1.807 |
| ConfVAE | 55.20 | 59.43 | 1.238 | 1.142 | 22.96 | 14.05 | 1.829 | 1.816 |
| GeoMol* | 67.16 | 71.71 | 1.088 | 1.059 | - | - | - | - |
| ConfGF | 62.15 | 70.93 | 1.163 | 1.160 | 23.42 | 15.52 | 1.722 | 1.686 |
| GeoDiff | 89.13 | 97.88 | 0.863 | 0.853 | 61.47 | 64.55 | 1.171 | 1.123 |
| DMCG | **96.69** | **100.0** | **0.722** | 0.724 | - | - | - | - |
| RDKit | 60.19 | 64.28 | 1.219 | 1.133 | 69.23 | 87.63 | 1.113 | 0.963 |
| OMEGA | 81.64 | 97.25 | 0.851 | 0.771 | 77.18 | 96.15 | 0.951 | 0.854 |
| RMCF-R | 82.25 | 90.77 | 0.839 | 0.789 | **83.02** | **98.50** | **0.812** | **0.722** |
| RMCF-C | 87.12 | 96.26 | 0.749 | **0.709** | 82.01 | 95.91 | 0.835 | 0.754 |

* We follow the results reported by Zhu et al. [2022], Xu et al. [2022], which use the same data splits as us. GeoMol achieves higher scores in Ganea et al. [2021]'s data splits. COV-R: 82.43/95.10, MAT-R 0.862/0.837, COV-P 78.52/94.40, MAT-P 0.933/0.856

## 4.3 Results and Discussions

The main results are presented in Table 1. We generated conformations by sampling on the RMCF with two strategies: (1) directly sample a specific number of conformations (i.e., $S_g$), and (2) first generate a sufficient number of conformations (we generate 10,000 conformations in this work), then cluster them into $S_g$ clusters and sample one conformation from each cluster. The model performance associated with the above two strategies are named RMCF-R and RMCF-C, respectively. As shown in Table 1, RMCF significantly outperforms all other models in precision metrics which means RMCF can generate more accurate and high quality conformations. We also achieve comparable performance in recall metrics with DMCG [Zhu et al., 2022] model, which is the current state-of-the-art model on recall metrics. We think RMCF and DMCG improve the conformation generation in different aspects. Generally, modeling more freedom in the conformation generation process enables the model to generate diverse outputs (e.g., DMCG), which benefits the recall metric, while reducing modeling

freedom will help to generate conformations more accurately, leading to better precision metric (e.g., RMCF).

In addition, we find that the clustering-after-sampling strategy leads to a boost in the structural diversity, while keeping a good quality (i.e., energetically favorable conformations) of the generated conformations.

Since we employ a discretized treatment for the continuous dihedral angle as well as the fragment 3D conformation, the model performance will inevitably have some discrepancy between the predicted and ground truth conformations. To that end, we investigate the upper and lower bound of our model performance to have a better understanding of its predicting power, as shown in Table 2. Specifically, we discretize the ground truth conformations using the molecular segmentation algorithm to obtain the fragment configurations ("gold $\mathcal{X}_f$") and inter-fragment dihedral angles ("gold $\mathcal{X}_d$"), then evaluate the corresponding metrics. On the other hand, we randomly sample some $\mathcal{X}_f$ and $\mathcal{X}_d$, whose evaluation results should indicate the performance lower bound using our fragment representation. Surprisingly, we find that although using randomly sampled $\mathcal{X}_f$ and $\mathcal{X}_d$, we can still outperform some previous models. This is an advantage of using molecular fragment as building blocks for conformation generation, since we bypass the need to generate many insignificant variables which may degrade the model performance. We also learn that having accurate dihedral angle predictions (i.e., $\mathcal{X}_d$) is more significant than good fragment configuration predictions (i.e., $\mathcal{X}_f$). These experimental results support our claim that capturing a few significant DoF within a molecule is adequate for generating a diverse set of low-energy conformations, while other DoF could be safely ignored during modeling.

Table 2: The empirical upper and lower bound of RMCF performance on the **GEOM-Drugs** dataset.

| RMCF Settings | COV-R (%) ↑ Mean | Median | MAT-R (Å) ↓ Mean | Median | COV-P (%) ↑ Mean | Median | MAT-P (Å) ↓ Mean | Median |
|---|---|---|---|---|---|---|---|---|
| gold $\mathcal{X}_f$,gold $\mathcal{X}_d$ | 99.16 | 100.0 | 0.302 | 0.242 | 99.37 | 100.0 | 0.290 | 0.235 |
| rand $\mathcal{X}_f$,gold $\mathcal{X}_d$ | 96.74 | 100.0 | 0.511 | 0.464 | 88.93 | 100.0 | 0.634 | 0.561 |
| gold $\mathcal{X}_f$,rand $\mathcal{X}_d$ | 66.21 | 77.42 | 1.125 | 1.092 | 40.30 | 34.25 | 1.429 | 1.397 |
| rand $\mathcal{X}_f$,rand $\mathcal{X}_d$ | 63.60 | 72.22 | 1.156 | 1.135 | 37.68 | 31.25 | 1.463 | 1.429 |

\* $\mathcal{X}_f$ denotes the fragment configuration set, $\mathcal{X}_d$ denotes the inter-fragment dihedral angle set
\* "gold" refers to the ground truth distribution, "rand" refers to a randomly sampled distribution.

At last, we showcase the generated conformations of two example molecules in Figure 5. For each molecule, we take the first three predicted conformations, and align the non-rigid parts for visualization purposes. From Figure 5(a), we observe a diverse set of conformations mainly driven by the rotation of two single bonds without causing too much steric effect. For the amide bond regions, the model correctly predicts the corresponding dihedral angles to form planar conjugate systems. Meanwhile, for Figure 5(b) we see a large planar conjugate system on the left side and a cyclooctane ring on the right. Again, the model accurately captures the conjugate system and only makes variations in the cyclooctane conformations. Interestingly, the segmentation algorithm tends to only preserve the cyclic groups (e.g., benzyl and furan groups), and even cuts through the amide bonds and other acyclic conjugate systems. This behavior is a direct consequence of balancing the vocabulary size and the variety of local chemical environment.

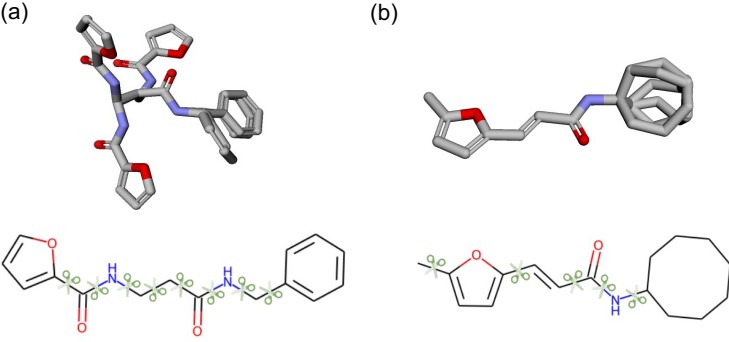

Figure 5: The first three generated conformations of two example molecules. The upper panel shows the 3D atomic arrangement, where the non-rigid fragments are aligned to help visualization. The lower panel shows where the segmentation has been made for each molecule, as indicated by the scissors.

We show that our model performs reasonable segmentation of drug-like molecules into small functional groups with limited internal DoF, and could correctly predict planar conjugate systems. Therefore, we believe that our generative process is essentially sampling from the local-minima of the learned potential energy surface by RMCF, where those non-essential DoF which contribute to local structural perturbations are omitted in our modeling framework. We argue that both the data-driven segmentation and MRF modeling of the joint probability distribution are essential for the success of our model.

## 5  Conclusion

We introduce RMCF, a novel framework for 3D molecular conformation generation. Our model is physics-motivated, with the central idea to effectively model the joint probability distribution of governing dynamical modes in a reduced conformation space to achieve energetically favorable conformation generation. Experimental results show that RMCF outperforms state-of-the-art models on the GEOM-Drugs dataset to predict a diverse set of conformations located at distinct local minima of the corresponding molecular potential energy surface. Our methodology can be naturally extended to larger biomolecular systems, e.g., proteins, whose conformation prediction is a significant topic in the biological research community. We will address this challenge in our future work.

## Acknowledgements

We would like to thank the anonymous reviewers for their constructive comments. Hao Zhou is the corresponding author. This work is jointly supported by Vanke Special Fund for Public Health and Health Discipline Development, Tsinghua University (NO.20221080053), Guoqiang Research Institute General Project, Tsinghua University (No. 2021GQG1012) and National Natural Science Foundation of China (No. 62076068).

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
