# Appendix

## A   Visualizations

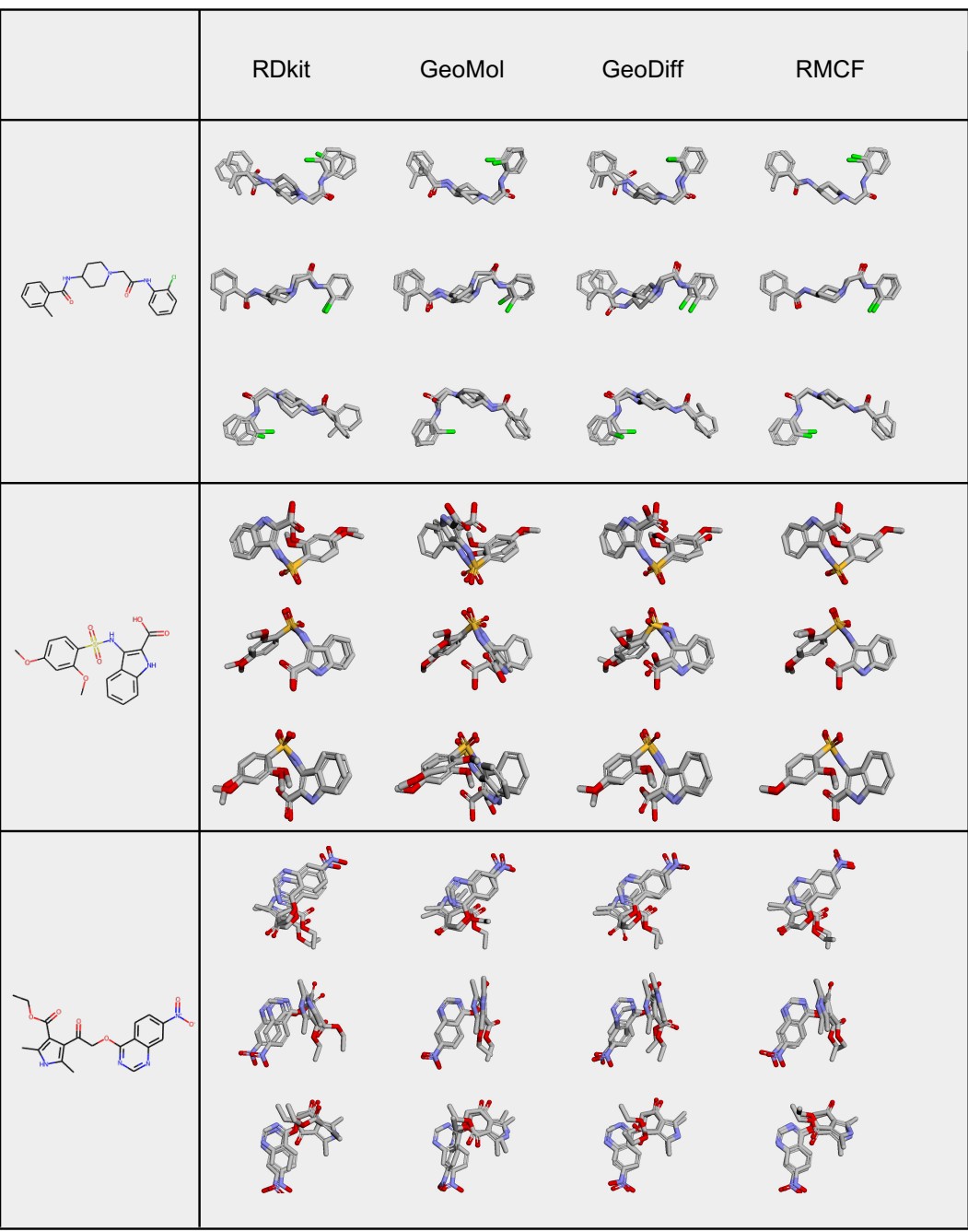

Figure 6: Examples of generated molecules from GEOM-Drugs dataset. For every model and molecule, we show three ground truths and the best-aligned conformations. Our model RMCF fits best with the references.

## B  Graph Dynamic Programming Algorithm

---

**Algorithm 1** Graph Dynamic Programming for Searching the Best Partition

---

**Require:** the fragments type collection $\mathcal{F}$, the molecular graph $G_m$
**Ensure:** the best molecular graph partition $\mathcal{P}^*$
  **function** DoF($f$)
    **return** $\frac{1}{|\mathcal{C}_f|}\sum_{\mathcal{C}_f}\mathrm{RMSD}(\mathcal{C}_f,\overline{\mathcal{C}_f})^2$
  **end function**
  **function** PARTITION($G$)
    $s \leftarrow 100, \mathcal{P} \leftarrow \{\}$
    **for** $f \in \mathcal{F}$ **do**
      **if** IsSubstructure($G, f$) **then**               ▷ Determine if $f$ is a substructure of $G$
        $s_m \leftarrow 0, z \leftarrow 1, \mathcal{P}_m \leftarrow \{f\}$
        $F_r \leftarrow$ REMOVE($G, f$)      ▷ Cut the $f$ and split $G$ into fragments set $F_r$ without $f$
        **for** $f \in F_r$ **do**
          $dof, \mathcal{P}_f \leftarrow$ PARTITION($f$)
          $s_m \leftarrow s + dof * |\mathcal{P}_f|, z \leftarrow z + |\mathcal{P}_f|, \mathcal{P}_m \leftarrow \mathcal{P}_m \cup \mathcal{P}_f$
        **end for**
        $s_m \leftarrow s_m/z$
      **end if**
      **if** $s_m < s$ **then**
        $s \leftarrow s_m, \mathcal{P} \leftarrow \mathcal{P}_m$
      **end if**
      **if** $\mathcal{P} = \{\}$ **then**
        **return** DoF($G$), $\{G\}$
      **end if**
    **end for**
  **end function**
  $dof, \mathcal{P}^* \leftarrow$ PARTITION($G_m$)
  **return** $\mathcal{P}^*$

---

## C  Sampling Algorithm

---

**Algorithm 2** Sampling Molecular Conformations From RMCF

---

**Require:** an RMCF $\mathcal{G}$, the number of conformations $N_C$, the number of sampling iterations $N_S$
**Ensure:** a set $S_C$ containing $N_C$ conformations
  $S \leftarrow \{\}$
  initialize a configuration $\mathcal{X}$ randomly
  **for** $i \in [1, 2, \cdots, N_S]$ **do**
    **for** $n \in F \cup D$ **do**
      $U \leftarrow \{u | (n, u) \in E\}$               ▷ $x_u$ is fixed in this iteration
      $P \leftarrow$ softmax $\left\{\sum_{u \in U} \mathbf{E}_u[x_u]\mathbf{W}_{un}\mathbf{E}_n^{\mathrm{T}} + \mathrm{out}(\mathbf{h}_n)\right\}$    ▷ the value can be pre-computed
      $x_n \sim \mathrm{Cat}(|n|, P)$
    **end for**
    add $\mathcal{X}$ to $S$
  **end for**
  compute the pairwise distance $K$ with Eq. 11
  $S_C \leftarrow$ run clustering with $k = N_C$ over $S$ based on $K$
  **return** $S_C$

---

# D QM9 Results

Table 3: Results on the **GEOM-QM9** dataset, without FF optimization.

| Models | COV-R (%) ↑ | | MAT-R (Å) ↓ | | COV-P (%) ↑ | | MAT-P (Å) ↓ | |
|---|---|---|---|---|---|---|---|---|
| | Mean | Median | Mean | Median | Mean | Median | Mean | Median |
| CVGAE | 0.09 | 0.00 | 1.671 | 1.609 | - | - | - | - |
| GraphDG | 73.33 | 84.21 | 0.425 | 0.397 | 43.90 | 35.33 | 0.581 | 0.582 |
| CGCF | 78.05 | 82.48 | 0.422 | 0.390 | 36.49 | 33.57 | 0.662 | 0.643 |
| ConfVAE | 77.84 | 88.20 | 0.415 | 0.374 | 38.02 | 34.67 | 0.622 | 0.609 |
| GeoMol* | 71.26 | 72.00 | 0.373 | 0.373 | - | - | - | - |
| ConfGF | 88.49 | 94.31 | 0.267 | 0.269 | 46.43 | 43.41 | 0.522 | 0.512 |
| GeoDiff | 90.07 | 93.39 | 0.209 | **0.199** | 52.79 | 50.29 | 0.445 | 0.440 |
| DMCG | **96.34** | **99.53** | **0.207** | 0.200 | - | - | - | - |
| RMCF-R | 55.86 | 56.67 | 0.433 | 0.414 | **84.86** | **96.51** | **0.260** | **0.227** |
| RMCF-C | 76.22 | 85.50 | 0.320 | 0.289 | 83.10 | 91.44 | 0.283 | 0.261 |

\* We follow the results reported by Zhu et al. [2022], Xu et al. [2022], which use the same data splits as us. GeoMol achieves higher scores in Ganea et al. [2021]'s data splits. COV-R: 91.52/100.0, MAT-R 0.225/0.193, COV-P 86.71/100.0, MAT-P 0.270/0.241

# E Hyperparamters

Table 4: Hyperparameter choices of RMCF and training phase

| Hyperparameters | Values |
|---|---|
| **Message Passing Neural Network** | |
| Number of MPNN Layers $N$ | 6 |
| Dimension of Embeddings $d_{embed}$ | 320 |
| Dimension of Hiddens $d_h$ | 320 |
| Dimension of Feed-forward Layers $d_{ff}$ | 1280 |
| Dropout Rate $P_{drop}$ | 0.1 |
| **Markov Random Fields Layer** | |
| Number of States $|f|$ or $|g|$ | 72 |
| Dimension of Hiddens $\mathbb{D}$ | 320 |
| **Training** | |
| Batch Size | 256 |
| Learning Rate | $5 \times 10^{-4}$ |
| Max Training Steps | $1.2 \times 10^6$ |
| Optimizer | Adam |
| Learning Rate Scheduler | Linear |

We conducted our experiments on 8 A100 GPUs and took about 80 hours to train RMCF on GEOM-Drugs.