# OpenReview forum: "Regularized Molecular Conformation Fields"
_NeurIPS.cc/2022/Conference — NeurIPS 2022 Accept_

### Official Review · Reviewer_QFQe · 2022-07-11

**Rating:** 6
**Confidence:** 4
**Soundness:** 3 good
**Presentation:** 3 good
**Contribution:** 4 excellent

**Summary:**

This work provides a way to generate small molecule conformations from their molecular graph. The key idea is to find an appropriate fragmentation of a set of small molecules and to learn a Markov Random Field for the joint probability distribution of small molecular fragments. The paper demonstrates the model on the GEOM-Drugs dataset and compares to previously published baselines.


**Questions:**


The presentation of the segmentation algorithm is unclear so I'll focus most of my questions on that topic.  In particular: 1) what is the clustering algorithm used and what is the way to decide the initial number of 3D clusters?  2) What aspect of equation (2) is optimized, i.e. how is the set of molecules re-fragmented (or is it not?) to obtain an optimal solution.  I did not spend much time, but from a quick search of the codebase, I could not find what part of the provided code performs this segmentation optimization.  3) The language in the paper makes me concerned that the authors might have accidentally used the full molecule set (including the validation split) during the construction of the 3D fragments, therefore possibly having some minor leakage.  If that is not the case, please fully clarify the description of this process and point to the relevant code.  4) How does the code deal with out-of-sample fragments?  If the fragmentation of a validation molecule leads to a piece that is not existing in the training set, what happens then?! 5) When there are multiple options, how are the atoms selected that perform the connecting pieces between fragments (in order to reduce the problem to only dihedrals) and does the selection influence the results at all? 6) would it be possible to avoid binning the angles and what if any would be the disadvantages of such an approach?  7) Finally, how do the authors address possible steric clashes from the generated conformations, or does the markov field somehow avoid from ever proposing such a clash even in reasonably complicated small molecules, e.g. a large macrocycle?


Although the following would not be necessary for acceptance of this work, it would be nice if the authors showed the behavior of their model for the dataset in Table 4 as a function of the cutoff parameter delta, if they clarified how they selected the examples for Figure 5, and if they could include a brief comparison with the torsional-diffusion model, whose preprint is here:

https://arxiv.org/abs/2206.01729



**Limitations:**


The authors did not adequately address any limitations of this work. Perhaps the authors could address how this method would scale to macromolecules, to molecules with fragments outside the training set, or to macrocycles.  Would it be possible to predict the structure of macrocycles if those were outside of a training set?


**Strengths And Weaknesses:**


The key idea behind this work is sound and original. The presentation occasionally leaves some open questions (see below).  Although there is some code that allows for the reproducibility of this work and would add to the clarity of this method, I haven't verified if the code actually works as advertised. The problem that this work addresses is significant enough so it warrants some additional consideration in my opinion, despite the possible problems with this first draft, some of which I highlight below.

---

> ### Author Response · Authors · 2022-08-02
> **Response to Reviewer QFQe**
>
>
>
> **Q1: What is the clustering algorithm used and what is the way to decide the initial number of 3D clusters?**
>
> A1: We apologize for missing this important detail. We use K-medoids algorithm for clustering and use the Silhouette Coefficient metric to choose the best the number of cluster k. K-medoids clustering algorithm guarantees the center of clusters are actual data points (instead of some averaged centers). We calculate the root-mean-square deviation (RMSD) of two aligned fragment conformations for clustering, which takes rotation and translation invariance into consideration. We adjust the number of cluster centers k from 1 to 10, and choose the optimal hyper-parameter k with the highest Silhouette Coefficient value. Please note that the clustering algorithm is applied to each molecule based on its 3D conformations, where the optimal k value is also molecule-dependent.
>
> **Q2: What aspect of equation (2) is optimized, i.e. how is the set of molecules re-fragmented (or is it not?) to obtain an optimal solution.**
>
> A2: Eq(2) is a dynamic programming problem. For each type of molecule, the final segmentation is deterministic and will not be optimized in subsequent steps and the molecules are no need to re-fragment several times. Please refer to our general response -> Q2 for more details about the implementation.
>
> **Q3: If authors might have accidentally used the full molecule set (including the validation split) during the construction of the 3D fragments?**
>
> A3: We removed the molecules from the validation and test set from the GEOM-Drugs dataset and used the remaining molecules to construct the vocabulary. We also measured the built-in deviation in Table 2. The corresponding code to preprocess vocabulary from scratch has been included in the updated code. In addition, we also directly provide the preprocessed 3D fragment vocabulary to help with reproduction and other follow-up work.
>
> **Q4: How to deal with out-of-sample fragments?**
>
> A4: We constructed our vocabulary on the full GEOM-Drugs dataset (removed the molecules from validation and test set) and we didn't meet the OOV problem.  We have out-of-training-set fragments but this does not cause errors. We reserved UNK tokens for the model and will use Rdkit software to generate the conformations of OOV fragments.
>
> **Q5: When there are multiple options, how are the atoms selected that perform the connecting pieces between fragments?**
>
>  A5: For each fragment species, we pre-define a fixed set of interface atoms (labelled by the fragment canonical SMILES atom ID) to ensure that we always use the same set of atoms to calculate the dihedral angle between fragments.
>
> **Q6: Would it be possible to avoid binning the angles and what if any would be the disadvantages of such an approach?**
>
> A6:
>
> 1. Under the Markov Random Fields Framework, since choice of 3D fragments is discrete, it will be a little bit hard to model the binary between continuous angle values and discrete fragment states, which leads to difficulties in training, inference, and sampling.
> 2. We understand that predicting continuous dihedral angles is more physically meaningful than using discretized values. However, the central idea behind this work is to effectively sample from a low-dimensional potential energy surface. To that end, we assume that small variations in dihedral angles will lead to near-degenerate conformations with similar energy, which could be treated as similar conformations. Therefore, using discretized dihedral angle values could reduce the dimension of the entire modeling space, hence helping our RMCF model to better capture a diverse set of low energy conformations from the reduced potential energy surface.
> 3. Notably, the discretization of 3D molecular fragment conformations introduced in our work bears a resemblance to our treatment on the dihedral angles, since the 3D fragment conformations are also continuous variables. The main purpose of our discretized treatment is to restrict the dimension of the PES spanned by fragment conformations and inter-fragment dihedral angles to facilitate generating (meta)stable conformations through RMCF.
>
> **Q7: How do the authors address possible steric clashes from the generated conformations?**
>
> A7: Thanks for your insightful question. Yes, we agree that the Markov random field explicitly modeled the fragment and dihedral dependencies which somehow avoid steric clashes.  As for larger systems, it depends on the long-range dependency modeling ability of the models. We plan to explore possible solutions to the steric clash problem in future work.

---

> > ### Comment · Area_Chair_7k5S · 2022-08-07
> > **AC request**
> >
> > Dear Reviewer QFQe,
> >
> > Please let authors know if they have addressed your concerns ASAP. The time to ask for further clarifications is running out.
> >
> > Thanks,
> > AC

---

> > ### Author Response · Authors · 2022-08-08
> > **Response to some additional questions**
> >
> > **Q8:   Clarify how to select the examples for Figure 5, and   briefly compare with the torsional-diffusion model.**
> >
> > A8:
> > 1、These two interesting cases in Figure 5 are selected according to two ideas:
> > (a) fragments conformation almost unchanged and  dihedral angles diverse,
> > (b) dihedral angles almost unchanged and (part of) fragments conformation diverse.
> > For example (a), we observe a diverse set of conformations mainly driven by the rotation of two single bonds.
> > For example (b), the model accurately captures the conjugate system (with fixed dihedral angles ) and only makes variations in the cyclooctane conformations, which is in accordance with the laws of chemistry.
> > 2、We greatly appreciate the kind reminder of the torsional-diffusion (named TorDiff later) paper [A], which has insightful perspective, rigorous proof, solid experiments, and good performance. We compared our model RMCF with the torsional-diffusion model in four aspects:
> > - Motivation: Both RMCF and TorDiff aim to restrict the DoF (degree-of-freedom) of molecule conformation. RMCF reaches this goal by graph dynamic programming, while TorDiff utilizes fast cheminformatics methods (e.g., RDKit's ETKDG).
> > - Modeling: Markov random field in RMCF has a similar form to Boltzmann distribution in TorDiff. They are energy-based models built over dihedral angles to score a conformation. In addition, RMCF takes into consideration the flexibility inside a fragment.
> > - Training: A significant difference lies in the training process -- RMCF uses piecewise training to approximately fit the Markov Random Field, while TorDiff computes the likelihood through the diffusion process and trains as a score-based model.
> > - Inference & Sampling: Since MCMC is a common strategy for sampling from Boltzmann distributions, RMCF and TorDiff have adopted similar methodologies. Besides, TorDiff employs Boltzmann generators for fast and accurate sampling, and RMCF applies loopy belief propagation for fast maximum-a-posterior inference. The Boltzmann generator is novel and the result is impressive, compared to vanilla MCMC methods.
> >
> > [A] Torsional Diffusion for Molecular Conformer Generation

---

> ### Author Response · Authors · 2022-08-04
> **Reply**
>
> Dear reviewer,
>
> Did our response and the updated manuscript (Section 3 rewritten to be more clear) address your concerns?
>
> Best regards,

---

### Official Review · Reviewer_u5WS · 2022-07-11

**Rating:** 7
**Confidence:** 4
**Soundness:** 4 excellent
**Presentation:** 4 excellent
**Contribution:** 3 good

**Summary:**

This paper proposes a novel fragment-based model for molecular conformation generation. The authors develop a data-driven molecular segmentation algorithm to break each molecule into several fragments. The model learns the joint probability distribution of fragment configurations and dihedral angles between fragments. The proposed method outperforms existing baselines with a clear margin.

**Questions:**

Questions:

- If I understand it correctly, the BRICS segmentation will give a unique segmentation for each molecule. It seems only clustering is needed after that, but where the "argmin P"  in Equation (2) comes from? Also, how do you determine the number of clustering centers?
- The segmentation algorithm returns a vocabulary of 9k 2D fragments and 30k 3D fragments. How is the frequency of each fragment? Since the number of modes is still very large, I'm curious about whether the model will have some difficulties in learning the long tail modes.
- Is it hard to train the model without discretizing the continuous angle values into bins?

Missing reference:
Besides EGNN and GeoDiff, [1] is also an equivariant network which directly learns 3D coordinates of atoms.

[1] Guan, Jiaqi, Wesley Wei Qian, Wei-Ying Ma, Jianzhu Ma, and Jian Peng. "Energy-Inspired Molecular Conformation Optimization." In International Conference on Learning Representations. 2021.

**Limitations:**

The authors didn't discuss limitations. There is not negative social impact.

**Strengths And Weaknesses:**

Strengths:
- The proposed fragment-based model for molecular conformation generation is novel.
- The paper is well-written and easy to follow.
- The proposed method shows great empirical performance.

Weaknesses:
- I have some concerns about the segmentation algorithms. See details in Questions.

---

> ### Author Response · Authors · 2022-08-02
> **Response to Reviewer u5WS**
>
>
>
> **Q1: Where does the "argmin P" in Equation (2) come from?**
>
> A1:  There are numerous ways to partition a molecule into multiple 2D fragments. Therefore, with the goal to effectively model the potential energy surface in a reduced molecular conformation space, we employ graph dynamic programming to obtain the segmentation strategy with the lowest flexibility inside the fragments. The dynamic programming objective described by Eq(2) involves searching for a solution P to partition a molecule into fragments with the lowest total intra-fragment degrees of freedom (DoF). We define the DoF of each molecular fragment as the maximum RMSD between all possible pairs in the fragment conformation vocabulary V(f). We rewrote the vocabulary construction section to help the readers better understand our processing pipeline.
>
> **Q2: how do you determine the number of clustering centers**
>
> A2: We use K-medoids algorithm for clustering and use the Silhouette Coefficient metric to choose the best the number of cluster k. K-medoids clustering algorithm guarantees the center of clusters are actual data points (instead of some averaged centers). We calculate the root-mean-square deviation (RMSD) of two aligned fragment conformations for clustering, which takes rotation and translation invariance into consideration. We adjust the number of cluster centers k from 1 to 10, and choose the optimal hyper-parameter k with the highest Silhouette Coefficient value. Please note that the clustering algorithm is applied to each molecule based on its 3D conformations, where the optimal k value is also molecule-dependent. We apologize for not mentioning this in the original manuscript.
>
> **Q3: Whether the model will have some difficulties in learning the long tail modes since the number of modes is still very large?**
>
> A3: Although the size of the entire 3D vocabulary is over 30k, each fragment typically only has less than 10 3D conformations. For each fragment, we mask all conformations of other types of fragment  during training and inference, which alleviates the long tail problem.
>
> **Q4: Is it hard to train the model without discretizing the continuous angle values into bins?**
>
> A4: Under the Markov Random Fields framework, since the choice of 3D fragments is discrete, it will be a little bit hard to model the binary interactions between continuous angle values and discrete fragment states, which will then lead to difficulties in training, inference, and sampling. Therefore, we also discretized dihedral angles in a similar manner to fragment conformations.
>
> **Q5: Missing Reference [1]**
>
> A5: Thanks for your suggestion. We have cited this work in our revised manuscript.

---

> > ### Comment · Area_Chair_7k5S · 2022-08-07
> > **AC comment**
> >
> > Dear Reviewer u5WS,
> >
> > Please let authors know if they have addressed your concerns ASAP. The time to ask for further clarifications is running out.
> >
> > Thanks,
> > AC

---

> ### Author Response · Authors · 2022-08-04
> **Reply**
>
> Dear reviewer,
>
> Did our response and the updated manuscript (Section 3 rewritten to be more clear) address your concerns?
>
> Best regards,

---

> > ### Comment · Reviewer_u5WS · 2022-08-07
> > **Thank you for the responses**
> >
> > Thank the authors for the responses. My concerns are well addressed. I remain in favor of accepting the paper.

---

### Official Review · Reviewer_JhnJ · 2022-07-13

**Rating:** 6
**Confidence:** 4
**Soundness:** 3 good
**Presentation:** 1 poor
**Contribution:** 3 good

**Summary:**

The authors propose a new method for 3D conformation generation. The method contains three steps:

(1) the authors use dynamic programming to segment a molecule into several fragments, where the objective is in Eqn.(2);

(2) Convert the generation problem into a classification problem: Given the fragments, predict the conformation and dihedral angles;

(3) Assemble the intermediate results to the conformation of the molecule.

The method achieves promising results several baselines.

**Questions:**

1. What is the best choice of $|\mathcal{V}(f)|$? Any ablation study?

2. I'd like to know more about the fragments you obtained. How many atoms of each atom (average pm std)? If the conformation of a fragment does not lie in a plain, how to determine the dihedral angles?


**Limitations:**

Yes

**Strengths And Weaknesses:**

## Originality:

The general idea is novel, and make sense. The idea is also similar to how to search the docking pose in molecular docking. The authors should comment on this.

## Significance:

1. The paper miss a reference [A1]. [A1] achieves better results on COV-R, and the mean of MAT-R.

[A1] https://arxiv.org/pdf/2202.01356.pdf


## Clarify:

The paper is hard to follow. This is my biggest concern of this paper.

1. A question to be confirmed: Given a fragment $f$, it should has its own V(f). The way to obtain V(f) is that: first, find the conformations of fragment $f$ in all molecules, and clustering them using some algorithm to obtain |V(f)| clusters. The centroid of each cluster is an element in V(f). Am I right? Do you take roto-translation invariant of the fragment conformations into consideration?


2. Line 134: "Then, we infer a configuration set $\mathcal{X}$ including the fragment state and the dihedral state": What do you mean by state? This is very important and should be explained clearly.

3. It is better to use some examples and figures to show what is $\mathcal{X}_f$, $\mathcal{X}_d$ and $\mathcal{V}_f$. Currently, it is hard for me to get it.

4. Eqn.(2): I did not get a clear solution about how to solve it.

---

> ### Author Response · Authors · 2022-08-02
> **Response to Reviewer JhnJ**
>
> **Q1: The idea is also similar to how to search for the docking pose in molecular docking.**
>
> A1: Thank you for the suggestion. We have added this point in Introduction to support the rationality of our approach in the revised version.
>
> **Q2: Miss a Reference [A1]**
>
> A2: Thank you for your suggestion. We have added it into the revised manuscript and compared the model with ours.
>
> **Q3: A question to be confirmed: Given a fragment f, it should has its own V(f). The way to obtain V(f) is that: first, find the conformations of fragment f in all molecules, and clustering them using some algorithm to obtain |V(f)| clusters. The centroid of each cluster is an element in V(f). Am I right? Do you take roto-translation invariant of the fragment conformations into consideration?**
>
> A3: Yes, you are right. We use K-Medoids clustering algorithm to guarantee the centroid of each cluster is an element. We use the RMSD as the distance between elements to address the roto-translation invariant issue. We apologize for missing the discussion on |V(f)|. For each fragment, we  adjust the number of  cluster centers k, from 1 to 10. We use the Silhouette Coefficient [a] to measure the goodness of these clusters and choose k with the largest Silhouette Coefficient value as the |V(f)|. Relevant discussions have been added to the revised manuscript.
>
> Reference:  [a] Silhouettes: a graphical aid to the interpretation and validation of cluster analysis
>
> **Q4: Line 134: "Then, we infer a configuration set X including the fragment state and the dihedral state": What do you mean by state? This is very important and should be explained clearly. It is better to use examples and figures to show what is Xf, Xd and Vf.**
>
> A4: Thank you for constructive suggestions. The molecular conformation consists of the choice of 3D conformation（**fragment states**）of each fragment and the dihedral angles (**dihedral states**) between the fragments. We will avoid using the confusing term "states", and use "configurations" in the revised manuscript. We also rewrote Section 3 with more examples and figures to explain the symbols and convey our main ideas.
>
> **Q5:  More details about Eq(2) and graph dynamic programming.**
>
> A5:  Please refer to our overall review response -> common question Q2 for more details.
>
> **Q6:  How many atoms are in each fragment? If the conformation of a fragment does not lie in a plain, how to determine the dihedral angles?**
>
> A6:
>
> 1. The number of atoms of non-ring fragments: mean 3.42, std 1.76.
>
> 2. The number of atoms of ring-containing fragments: mean 7.44, std 2.86.
>
> 3. As shown in Figure 3 and Eq(7), the dihedral angle calculation between two fragments involves 4 atoms and 3 bonds (with 1 common bond connecting two fragments).  The dihedral angle only describes the angle between the two planes intersecting at the common bond, which has nothing to do with other fragment atoms. Therefore, as long as we have a pre-defined set of interface atoms for each fragment, we can always calculate the dihedral angle between fragments using their atomic coordinates (taking roto-translation invariance into consideration).

---

> > ### Comment · Area_Chair_7k5S · 2022-08-07
> > **AC comment**
> >
> > Reviewer JhnJ,
> >
> > Please let authors know if they addressed your concerns.

---

> > ### Comment · Reviewer_JhnJ · 2022-08-08
> > **Thanks for your response.**
> >
> > Thanks for your response.
> >
> > 1. Line 18： “We adjust the number of cluster centers k from 1 to 10”： Do you tune the number of cluster centers for each fragment f, or use the same number of clusters for all fragments?
> >
> > 2. What does the "Remove" function mean in Appendix D?
> >
> > 3. Results are not as good as reference [A1], and the authors did not provide essential discussion.
> >
> > ## Additional comments
> >
> > 1. The GeoMol paper uses a different data partition as the ConfGF, GeoDiff. Seems that you simply use the results from GeoMol paper, which is not correct.

---

> > > ### Author Response · Authors · 2022-08-08
> > > **Thanks a lot for your further comments and response to your questions**
> > >
> > > Thanks a lot for your further comments. We answer your questions as follows. If you have remaining questions about this paper, we are happy to have more discussions.
> > >
> > > **Q1: Do you tune the number of cluster centers for each fragment f, or use the same number of clusters for all fragments?**
> > >
> > > A1: We tune the number of cluster centers for each fragment f. For each fragment, we perform  clustering several times, by enumerating k from 1 to 10. Then we use the Silhouette Coefficient [C] to measure the effectiveness of these clusters and choose the optimal hyper-parameter k with clusters having the highest Silhouette Coefficient value.
> > >
> > > **Q2: What does the "Remove" function mean in Appendix D?**
> > >
> > > A2: The "Remove" function in Remove(G, f) means breaking the original molecule G by removing fragement f from G, and returning remaining fragements. Specifically, you can refer to the function "remove_core" in segmentor/chain_spliter.py. Thanks for your comment and we have made it clear in the paper.
> > >
> > > **Q3: providing essential discussions with reference [A1]**
> > >
> > > A3: Thanks for your kind notes and we have added more discussions comparing RMCF and DMCG [A1]. We have also updated the manuscript.
> > >
> > > Specifically, [A1] presents a model called DMCG which is  invariant to roto-translation of conformation coordinates and permutation of symmetric atoms in molecules.  They use a VAE-based model  and design a dedicated loss function as the minimal distance between two sets of coordinates after any permutation and roto-translation. They give strong experimental results  under different settings and achieve current SOTA recall metrics. We compare our model with DMCG in two aspects:
> > >
> > > 1、DMCG  calculates coordinates on atoms which is straightforward and gives more freedom,  while our model tries to reduce the degree of freedom (DoF) and handles  the conformation as a combination of fragments and dihedral angles.
> > >
> > > 2、DMCG use VAE framework to generate diverse conformations which are sampled in continuous space  while our model uses Gibbs sampling  on Markov Random Fields to sample in discrete space.
> > >
> > > - We think the two different models improve the conformation generation in different aspects. Generally, modeling more freedom in the conformation generation process enables the model to generate diverse outputs (e.g., DMCG), which  benefits the recall metric, while reducing modeling freedom will help to generate conformations more accurately, leading to better precision metric (e.g., RMCF).
> > > - Such intuitions are in accord with our empirical results. As shown in Table 1, DMCG achieves  SOTA recall metrics while RMCF outperforms  other models on precision metrics by large margins.
> > >
> > > **Q4: The GeoMol paper uses a different data partition as the ConfGF, GeoDiff. Seems that you simply use the results from GeoMol paper, which is not correct.**
> > >
> > > A4: Thank you for your comment. Indeed, there are two versions of GeoMol's performance  (both of them are lower than RCMF on Geom-Drugs), the one reported in the GeoMol paper and the one reported in [A1][B] due to different data partition in their paper.  After considering comments from all reviewers, we decide to report the reproduced results from [A1][B] in the main table because we adopt the same data partition as [A1][B]. However, we will still  report scores from the original GeoMol paper in the table caption and add notes to make this clear.
> > >
> > > [A1] Direct Molecular Conformation Generation
> > >
> > > [B] GeoDiff: a Geometric Diffusion Model for Molecular Conformation Generation
> > >
> > > [C] Silhouettes: a graphical aid to the interpretation and validation of cluster analysis

---

> > > > ### Comment · Reviewer_JhnJ · 2022-08-09
> > > > **Thanks**
> > > >
> > > > I'm satisfied with the revision.
> > > >
> > > > ## Minor (you do not need to do it now, perhaps add them in the next version)
> > > > 1. Give a detailed analysis of your conformation fragments, and discuss with chemical experts to see whether you can get more insights.
> > > >
> > > > 2. How do you choose the cases in Figure 6? For what kind of molecules can your method generate better conformations? This is a hard work but with much values.
> > > >
> > > > 3. You can follow the Figure 7 of GeoMol paper and give an analysis w.r.t. to rotatable bonds.
> > > >
> > > > 4. Release the code and the pre-trained checkpoint.
> > > >
> > > >
> > > > In conclusion, I think this is a novel paper and I'd like to set my score as 6.

---

> ### Author Response · Authors · 2022-08-04
> **Reply**
>
> Dear reviewer,
>
> Did our response and the updated manuscript (Section 3 rewritten to be more clear) address your concerns?
>
> Best regards,

---

### Official Review · Reviewer_ckMd · 2022-07-15

**Rating:** 7
**Confidence:** 3
**Soundness:** 3 good
**Presentation:** 3 good
**Contribution:** 3 good

**Summary:**

Representation learning for chemistry and molecules in specific is an emerging area in deep learning. In this work, the authors propose a framework (and then a model) to generate low energy 3d conformations of molecules. The work proposes to segment a given molecule into  fragments (with limited internal flexibility), and then proposes to learn a joint probability distribution over the fragments using a Markov Random Field. As parameter estimation for the MRF is intractable, the authors provide an approximate learning strategy leveraging GNNs and Gibbs sampling to achieve their objective.

**Questions:**

1. Why not perform evaluation on Geom-QM9? Another dataset considered in the baseline models.
2. For me, it is not clear how you back propagate through the RMCF procedure. Please add detail
3. Similarly, please elaborate on the graph dynamic programming and also Eq (3). Specifically What does the GNN output? I was under the opinion it is the edge representations but then Eq(3) has a set on the left.
4. Also please address the concerns in the weaknesses section.

**Limitations:**

The authors have discussed the limitations of their work. For example:
1. Have discussed intractability of parameter estimation in Sec 3.4, and ways to counteract it.
2. In Sec 4.3, have discussed a few limitations of the segmentation framework

I do not see any potential negative societal impact of their work

**Strengths And Weaknesses:**

Strengths:
1. The idea to segment a molecule into fragments, and then use the framework to generate low energy. states is interesting.
2. The observation that even randomly sampled $\mathcal{X}_f, \mathcal{X}_d$ in section 4.3 can outperform some previous works is insightful

Weaknesses:
1. In my opinion, the writing is not precise and hard to understand. For e.g. a) in section 3.2, it is hard to decipher what is a sufficient number of 3D fragments? b) in sec 3.3 - how is the oracle configuration obtained?  c) in sec 3.5 the sampling procedure, it is not clear what the samples are? the fragments? 3d positions? and how detailed balance is satisfied d) Why are the side chains removed in 3.2, etc. Infact, i felt it was incredibly hard to read the entire of section 3.
2. Evaluation performed on a single dataset. Also the scores from other works appear to be incorrect (e.g. GeoMol scores are better than numbers presented here, etc) in addition to RMCF not outperforming other benchmarks.
3. The idea to fragment and compute graph representations have already been performed (e.g. especially energy based models [1]), which have not been cited. The idea to use GNNs to parametrize them has also been done in [1]

References:
1. Cotta, Leonardo, et al. "Unsupervised joint k-node graph representations with compositional energy-based models." Advances in Neural Information Processing Systems 33 (2020): 17536-17547.



Post Rebuttal Updates:

Thank you very much for the rebuttal and answering my questions on the dataset (QM9 has smaller molecules, so impacts performance), backprop and other questions I had raised. I have also read through the other reviews and the authors response. In light of all this, I am updating my score.

---

> ### Author Response · Authors · 2022-08-02
> **Response to Reviewer ckMd (Part 1)**
>
> Thank you for your constructive comments. Please see below for our response to your specific questions:
>
> **Q1: Why not perform evaluation on Geom-QM9? Another dataset considered in the baseline models.**
>
> A1: We performed additional experiments on GEOM-QM9, the results are tabulated below.
>
> | Models | COV-R Mean | COV-R Median |  MAT-R Mean |  MAT-R Median | COV-P Mean | COV-P Median | MAT-P Mean | MAT-P Median |
> | --- | --- | --- | --- | --- | --- | --- | --- | --- |
> | CVGAE | 0.09 | 0.00 | 1.671 | 1.609 | - | - | - | - |
> GraphDG | 73.33 | 84.21 | 0.425 | 0.397 | 43.90 | 35.33 | 0.581 | 0.582 |
> | CGCF | 78.05 | 82.48 | 0.422 | 0.390 | 36.49 | 33.57 | 0.662 | 0.643 |
> | ConfVAE | 77.84 | 88.20 | 0.415 | 0.374 | 38.02 | 34.67 | 0.622 | 0.609 |
> | GeoMol | 71.26 | 72.00 | 0.373 | 0.373 | - | - | - | - |
> | ConfGF | 88.49 | 94.31 | 0.267 | 0.269 | 46.43 | 43.41 | 0.522 | 0.512 |
> | GeoDiff | 90.07 | 93.39 | 0.209 | 0.199 | 52.79 | 50.29 | 0.445 | 0.440 |
> | RMCF-R | 55.86 | 56.67 | 0.433 | 0.414 | 84.86 | 96.51 | 0.260 | 0.227 |
> | RMCF-C | 76.22 | 85.50 | 0.320 | 0.289 | 83.10 | 91.44 | 0.283 | 0.261 |
>
> Our model achieves significantly better precision metrics and comparable recall metrics compared to other models. In general, molecules in the GEOM-QM9 dataset are relatively small (i.e., with up to 9 heavy atoms). Therefore, the error associated with discretizing continuous fragment conformation and inter-fragment dihedral angles will be magnified, hence limiting the generative power of our model. To that end, we originally decided to use the GEOM-Drugs dataset with larger molecules to demonstrate the advantage of RMCF.
>
> For a fair comparison with other models, we have added the relevant results and discussions on GEOM-QM9 to the revised manuscript .
>
>
>
> **Q2: For me, it is not clear how you back propagate through the RMCF procedure. Please add detail.**
>
> A2: We encode the 2D fragments and dihedral angles using a GNN to learn contextual representations of a molecule, as shown in Eq(3). The MRF model uses these learned representations and further models their joint probability distribution to obtain the loss function defined in Section 3.3. Then, we could back propagate from the loss function to update the fragment and dihedral angle embeddings. Please refer to Section 3.3 of the revised manuscript for more details.
>
>
> **Q3: Similarly, please elaborate on the graph dynamic programming and also Eq(3). Specifically What does the GNN output? I was under the opinion it is the edge representations but then Eq(3) has a set on the left.**
>
> A3:
>
> 1. There are numerous ways to partition a molecule into multiple 2D fragments. Therefore, with the goal to effectively model the potential energy surface in a reduced molecular conformation space, we employ graph dynamic programming to obtain the segmentation strategy with the lowest flexibility inside the fragments. The dynamic programming objective described by Eq(2) involves searching for a solution P to partition a molecule into fragments with the lowest total intra-fragment degrees of freedom (DoF). We define the DoF of each molecular fragment as the maximum RMSD between all possible pairs in the fragment conformation vocabulary V(f). We rewrote the vocabulary construction section to help the readers better understand our processing pipeline.
>
> 1. Graph dynamic programming algorithm: First search for a substructure from the fragment collection, then split the molecule into several substructures to minimize the mean of DoF of all substructures. Every substructure is  also a nested sub-problem until the substructure can no longer be further split. We add the detailed algorithm in Appendix D.
>
> 1. The input of GNN includes embeddings of 2D molecular fragments and dihedral angles, while the output is contextual representations to be used by the MRF model. We have updated Section 3 with a more concrete description of our model.
>
>
>
> **Q4: What is a sufficient number of 3D fragments?**
>
> A4:  We sample 1000 3D fragments for clustering. Empirically, we find that the optimal number of cluster centers (k) determined by the Silhouette Coefficient metric will be stabilized. Therefore, we use 1000 as a sufficient number of 3D fragments in order to balance structural diversity and computational efficiency.
>
>
>
> **Q5: How is the oracle configuration obtained in Sec 3.3?**
>
> A5: The oracle configuration contains the oracle choice of fragment conformation and dihedral angles. For every fragment, we use the cluster center with the lowest RMSD to the ground truth conformation as the oracle conformation. The oracle dihedral angles are determined according to the ground truth dihedral angles after discretization (i.e., binning).

---

> > ### Author Response · Authors · 2022-08-02
> > **Response to Reviewer ckMd （Part2）**
> >
> > **Q6: In sec 3.5 the sampling procedure, it is not clear what the samples are? the fragments? 3d positions? and how detailed balance is satisfied.**
> >
> > A6: In our proposed work, the molecular conformation space is spanned by its fragment conformations and the inter-fragment dihedral angles. A specific sample consists of a particular 3D conformation for each fragment, and a choice of inter-fragment dihedral angle for each pair of connected fragments. We then assemble the fragments into a 3D conformation according to the predicted dihedral angles, as described in Section 3.5 in the revised manuscript. We pre-sampled a large number (e.g., 10,000) of samples to guarantee the balance is satisfied.
> >
> >
> >
> > **Q7: Why are the side chains removed in 3.2.**
> >
> > A7: We apologize for the confusion. Since BRICS preserves some sides chains on rings, this would lead to an oversized vocabulary due to combinatorial explosion. Therefore, we further split the side chains and rings, and both are added to our vocabulary (not removed). We modified the manuscript to clarify this point.
> >
> >
> >
> > **Q8: The performance of other works (e.g. GeoMol ) appear to be incorrect.**
> >
> > A8: Thanks for pointing this out. Since we used the same test set as that in GeoDiff, we followed their reported performance metrics using this test set. The original GeoMol paper used a different test set. For a fair comparison, we now use the metrics reported in the original GeoMol paper in our revised manuscript, and left a note about the test set we used.
> >
> >
> >
> > **Q9: Some ideas have already been performed in [1].**
> >
> > A9: Thanks for your suggestion. We cited this work in our revised manuscript.

---

> > > ### Comment · Area_Chair_7k5S · 2022-08-07
> > > **AC comment to reviewer ckMd**
> > >
> > > Dear Reviewer ckMd,
> > >
> > > Reading the review, rebuttal, main paper and appendix, it seems like the authors addressed most of the reviewer's questions (which were valid questions). Reviewer ckMd, reacting to the author's response ASAP is important to guarantee that any further concerns & doubts can still be addressed by the authors.
> > >
> > > Thanks,
> > > AC

---

> > > > ### Comment · Reviewer_ckMd · 2022-08-07
> > > > **Reply to authors rebuttal.**
> > > >
> > > > Dear Authors,
> > > >
> > > > Thank you very much for the rebuttal and answering my questions on the dataset (QM9 has smaller molecules, so impacts performance), backprop and other questions I had raised. I have also read through the other reviews and the authors response. In light of all this, I am updating my score.
> > > >
> > > > Best,
> > > > Reviewer

---

> ### Author Response · Authors · 2022-08-04
> **Reply**
>
> Dear reviewer,
>
> Did our response and  the updated manuscript (Section 3 rewritten to be more clear) address your concerns?
>
> Best regards,

---

### Author Response · Authors · 2022-08-02
**Response to all reviewers and area chair**


We appreciate your time and effort in reviewing our work, and thank you all for the constructive and critical suggestions on our proposed method.

We made the following two major changes to our revised manuscript (also highlighted in blue in the main text):

1. We rewrote the entire Section 3 to give the readers a clear presentation of the proposed RMCF model, with high-level ideas emphasized and technical details explained.

2. We performed benchmark experiments on the GEOM-QM9 dataset for a fair comparison with other models (e.g., GeoMol and GeoDiff), in Appendix C.

We also address some of the commonly raised questions from the reviewers:

**Q1: What is the clustering algorithm used and how to decide the cluster number k in Section 3.2?**

-  We use K-medoids algorithm for clustering and use the Silhouette Coefficient metric to choose the best the number of cluster k. K-medoids clustering algorithm guarantees the center of clusters are actual data points (instead of some averaged centers). We calculate the root-mean-square deviation (RMSD) of two aligned fragment conformations for clustering, which takes rotation and translation invariance into consideration. We adjust the number of cluster centers k from 1 to 10, and choose the optimal hyper-parameter k with the highest Silhouette Coefficient value. Please note that the clustering algorithm is applied to each molecule based on its 3D conformations, where the optimal k value is also molecule-dependent. We apologize for not mentioning this in the original manuscript. Relevant discussions have been added to the revised manuscript.


**Q2: Please provide more details about Equation 2, which defines the graph dynamic programming algorithm.**

- The objective described by Eq(2) involves searching for a solution P to partition a molecule into fragments with the lowest total intra-fragment degrees of freedom (DoF). We define the DoF of each molecular fragment as the maximum RMSD between all possible pairs in the fragment conformation vocabulary V(f). We rewrote the vocabulary construction section to help the readers better understand our processing pipeline.
- Graph dynamic programming algorithm: First search for a substructure from the fragment collection, then split the molecule into several substructures to minimize the mean of DoF of all substructures. Every substructure is  also a nested sub-problem until the substructure can no longer be further split. We add the detailed algorithm in Appendix D.


**Q3:  Why discretize the continuous dihedral angle values?**

- We understand that predicting continuous dihedral angles is more physically meaningful than using discretized values. However, the central idea behind this work is to effectively sample from a low-dimensional potential energy surface. To that end, we assume that small variations in dihedral angles will lead to near-degenerate conformations with similar energy, which could be treated as similar conformations. Therefore, using discretized dihedral angle values could reduce the dimension of the entire modeling space, hence helping our RMCF model to better capture a diverse set of low energy conformations from the reduced potential energy surface.
- Notably, the discretization of 3D molecular fragment conformations introduced in our work bears a resemblance to our treatment on the dihedral angles, since the 3D fragment conformations are also continuous variables. The main purpose of our discretized treatment is to restrict the dimension of the PES spanned by fragment conformations and inter-fragment dihedral angles to facilitate generating (meta)stable conformations through RMCF.

---

### Meta-Review · Area_Chair_7k5S · 2022-08-26

**Recommendation:** Accept
**Confidence:** Certain

**Metareview:**

This work introduces fragment-based data-driven model for molecular conformation generation. The method has a segmentation that breaks each molecule into several fragments and then learns a joint probability distribution over the fragment configurations and dihedral angles between fragments with a Markov Random Field, which enables the method to sample from different low-energy regions of a conformation space.

The idea is novel with convincing results. There were some initial questions about clarity and further discussions that were necessary but these have been mostly addressed during rebuttal, and reviewers have increased their scores once those were cleared out. Overall this is a solid paper and we recommend acceptance.

**Award:**

No

---

### Decision · Program_Chairs · 2022-09-14

Accept